# Non-canonical Wnt signalling regulates scarring in biliary disease via the planar cell polarity receptors

D.H. Wilson [1], E.J. Jarman [1], R.P. Mellin [1,14], M.L. Wilson[1], S.H. Waddell[1], P. Tsokkou [1], N.T. Younger [1], A. Raven[2], S.R. Bhalla[3], A.T.R. Noll[4], S.W. Olde Damink[4,5], F.G. Schaap[4,5], P. Chen[6], D.O. Bates[3,7], J.M. Banales[8], C.H. Dean [9], D.J. Henderson[10], O.J. Sansom [2,11], T.J. Kendall [12,13] & L. Boulter [1]*

The number of patients diagnosed with chronic bile duct disease is increasing and in most cases these diseases result in chronic ductular scarring, necessitating liver transplantation. The formation of ductular scaring affects liver function; however, scar-generating portal fibroblasts also provide important instructive signals to promote the proliferation and differentiation of biliary epithelial cells. Therefore, understanding whether we can reduce scar formation while maintaining a pro-regenerative microenvironment will be essential in developing treatments for biliary disease. Here, we describe how regenerating biliary epithelial cells express Wnt-Planar Cell Polarity signalling components following bile duct injury and promote the formation of ductular scars by upregulating pro-fibrogenic cytokines and positively regulating collagen-deposition. Inhibiting the production of Wnt-ligands reduces the amount of scar formed around the bile duct, without reducing the development of the pro-regenerative microenvironment required for ductular regeneration, demonstrating that scarring and regeneration can be uncoupled in adult biliary disease and regeneration.

[1] MRC Human Genetics Unit, Institute for Genetic and Molecular Medicine, Edinburgh, UK. [2] Cancer Research UK Beatson Institute, Glasgow, UK. [3] Cancer Biology, Division of Cancer and Stem Cells, School of Medicine, University of Nottingham, Centre for Cancer Science, Queen's Medical Centre, Nottingham, UK. [4] Department of Surgery, Maastricht University, Maastricht, The Netherlands. [5] Department of General, Visceral and Transplantation Surgery, RWTH University Hospital Aachen, Aachen, Germany. [6] Department of Cell Biology, Emory University School of Medicine, Atlanta, GA 30322, USA. [7] COMPARE University of Birmingham and University of Nottingham Midlands, Birmingham, UK. [8] Biodonostia HRI, CIBERehd, Ikerbasque, San Sebastian, Spain. [9] National Heart and Lung Institute, Imperial College London, London, UK. [10] Cardiovascular Research Centre, Institute of Genetic Medicine, Newcastle University, Newcastle, UK. [11] Institute of Cancer Sciences, University of Glasgow, Glasgow G61 1QH, UK. [12] University of Edinburgh Centre for Inflammation Research, Edinburgh, UK. [13] Edinburgh Pathology, University of Edinburgh, Edinburgh, UK. [14] Present address: Infectious Diseases and Immune Defence, The Walter and Eliza Hall Institute of Medical Research, Melbourne, Australia. *email: luke.boulter@ed.ac.uk

ntroduction: Biliary diseases, also known as cholangiopathies, account for approximately one-third of all adult liver disease and 70% of childhood liver disease[1]. Chronic damage to the bile duct typifies biliary disease and results in the proliferation of biliary epithelial cells (BECs) and the formation of a regenerative microenvironment populated by immune cells and fibroblasts[2]. These cells provide both instructive signals to BECs[3,4] and maintain tissue integrity by depositing scar tissue[5–7], thereby promoting bile duct regeneration. It remains unclear whether BECs communicate to the microenvironment to maintain it appropriately and regulate the amount of extracellular matrix (ECM) that is deposited by surrounding cells.

In a number of tissues, epithelial regeneration relies on the activation of Wnt-β-catenin signalling by locally produced Wnt ligands[8–10], and previous studies have concluded that BEC proliferation is β-catenin dependent[11,12]. Recent advances in both animal models[13] and single-cell RNA sequencing however, have challenged the central role for Wnt-β-catenin signalling in BEC proliferation and have suggested that the role of Wnt in these processes could be more complex than previously thought[14,15]. In addition to stabilising β-catenin, Wnt ligands integrate into a much larger signalling network, thereby activating a number of Wnt-dependent signalling cascades[16,17]. These, so-called, non-canonical Wnt signalling pathways are highly conserved from *Drosophila* through to vertebrates and utilise a broad range of cell surface receptors to activate diverse downstream processes[18–20]. Wnt-Planar Cell Polarity (Wnt-PCP) signalling represents one of these non-canonical pathways and is required for a number of morphogenic processes in the embryo;[21,22] moreover, Wnt-PCP signalling has been implicated in the pathogenesis of a number of adult diseases and cancers[23–25]. Whether Wnt-PCP signalling plays a role in bile duct disease and regeneration has not been determined.

Here, we demonstrate that Wnt ligands associated with non-canonical Wnt signalling, particularly Wnt5a[26], are upregulated in biliary injury. In this context, therapeutic inhibition of the Wnt signalling pathway, through the prevention of Wnt-ligand secretion, reduces the level of fibrosis deposited around proliferating BECs, without affecting BEC number. We then go on to demonstrate that Wnt ligands regulate this process through Planar Cell Polarity receptors that activate the JNK/c-JUN signalling pathway specifically in BECs. In turn, this Wnt-PCP signal promotes BEC crosstalk with portal fibroblasts and regulates the capacity of fibroblasts to synthesise collagen and form scar tissue. This study demonstrates how non-canonical Wnt signalling functions to regulate adult tissue scarring by integrating a number of cell types and offers a novel therapeutic target to treat biliary diseases in patients.

## Results

### Wnt-PCP signalling is activated during duct regeneration.
BEC proliferation is required during bile duct regeneration;[27] however, the role that Wnt signalling plays in this process remains controversial, with conflicting reports describing variable roles for Wnt-β-catenin[13,14,28]. Using tissue from patients with primary sclerosing cholangitis (PSC), a progressive human biliary disease in which BECs proliferate[29] and also two mouse models of BEC proliferation (thioacetamide, TAA or 3,5-diethoxycarbonyl-1,4-dihydrocollidine, DDC[30,31]) we sought to determine whether the Wnt-β-catenin pathway is activated in BECs. To do this, we assessed *Axin2* mRNA expression and the nuclear translocation of β-catenin in BECs. We failed to see that β-catenin translocates into the nucleus of BECs nor did we see the expected increase in *Axin2* expression in any of these contexts (Supplementary Fig. 1). Despite seeing no changes in these models, we have found that as

in many other systems[32,33] *Axin2* mRNA expression is responsive to changes in canonical Wnt signalling in BECs. In both mouse and human BECs, *Axin2* mRNA is increased following β-catenin stabilisation using a GSK3β inhibitor, CHIR99021, and decreases when β-catenin-dependent transcription is inhibited by PRI724[34] (Supplementary Fig. 1a). Therefore, our data suggest that whilst the Wnt-β-catenin pathway can be activated pharmacologically in BECs, activation of this pathway does not increase in BECs during bile duct regeneration. These data are in concordance with recent work showing that BECs do not express LGR proteins necessary for Wnt signalling potentiation[14,35], and that LRP-dependent Wnt signalling is dispensable for BEC organoid growth in vitro[36]. (We discuss these data in more detail in the Supplementary Discussion).

In addition to activating Wnt-β-catenin signalling, Wnt ligands also act via an alternative Wnt pathway known as Wnt-PCP signalling, which, through the activation of Rho-GTPases and JNK/c-JUN[19,37], promotes ductular formation in a number of embryonic contexts[21,38]. In liver tissue from patients with PSC, the number of BECs with phosphorylated JNK (phospho-JNK$^{T183/Y185}$) is significantly increased, and while c-JUN is expressed broadly within BECs, c-JUN phosphorylation (phospho-c-JUN$^{S73}$) is increased in PSC patients compared with those without disease (Fig. 1a, b), indicating that in ductular regeneration, the Wnt-PCP signalling pathway is likely activated.

Native liver explants from patients with PSC are severely fibrotic/cirrhotic and so represent end-stage disease. Therefore, it is not possible to determine whether Jnk/c-Jun signalling is dynamically regulated during disease progression. To characterise this, we evaluated the expression of phospho-Jnk$^{T183/Y185}$ and phospho-c-JUN$^{S73}$ during regeneration in two models of biliary injury and BEC proliferation and found that the number of phospho-JNK$^{T183/Y185}$ and phospho-c-JUN$^{S73}$ positive biliary cells is significantly increased compared with uninjured bile ducts (Fig. 1c, d). Furthermore, in isolated bile ducts from mice with biliary disease, matrix metalloproteinase-7 (*Mmp7*) and connective tissue growth factor (*Ctgf*), known c-JUN target genes[39], which are associated with biliary scarring[40,41], are also transcriptionally upregulated (Fig. 1e) and localise to BECs specifically (Fig. 1f, positive and negative RNAscope controls, supplementary Fig. 1h).

### Wnt–Jnk/c-Jun regulates biliary scar formation.
Previous studies have found that loss of Wnt-ligand secretion following Wnt-less (Wls) deletion alters biliary regeneration and fibrosis[13,42,43]. However, recent data suggesting that these ligands are not functioning though a Wnt-β-catenin pathway in BECs[13,14], led us to hypothesise that Wnt ligands activate Wnt-PCP signalling in bile ducts. To validate that Wnt ligands were upregulated in response to injury, we evaluated the mRNA expression of all Wnt ligands in isolated bile ducts from both the DDC and TAA models (Supplementary Fig. 2), thereby identifying a number of ligands that were transcriptionally overexpressed compared with healthy ducts. To resolve whether Wnt ligands are activating a Wnt-PCP signalling pathway during biliary repair, wild-type mice with biliary injury were given LGK974[44], a highly selective inhibitor of the MBOAT acyltransferase, Porcupine, which is required to add lipid modifications to Wnt ligands (LGK974 from hereon is referred to as Porcupine-i). Inhibition of Porcupine results in the global suppression of Wnt-ligand secretion and renders Wnt ligands unable to bind to the CRD domain of Frizzled. Our experimental strategy is summarised in Fig. 2a.

Following administration of Porcupine-i during biliary regeneration, isolated bile ducts had a reduced transcriptional expression of *Ctgf* and *Mmp7* (Fig. 2b). However, Wnt-β-catenin target genes

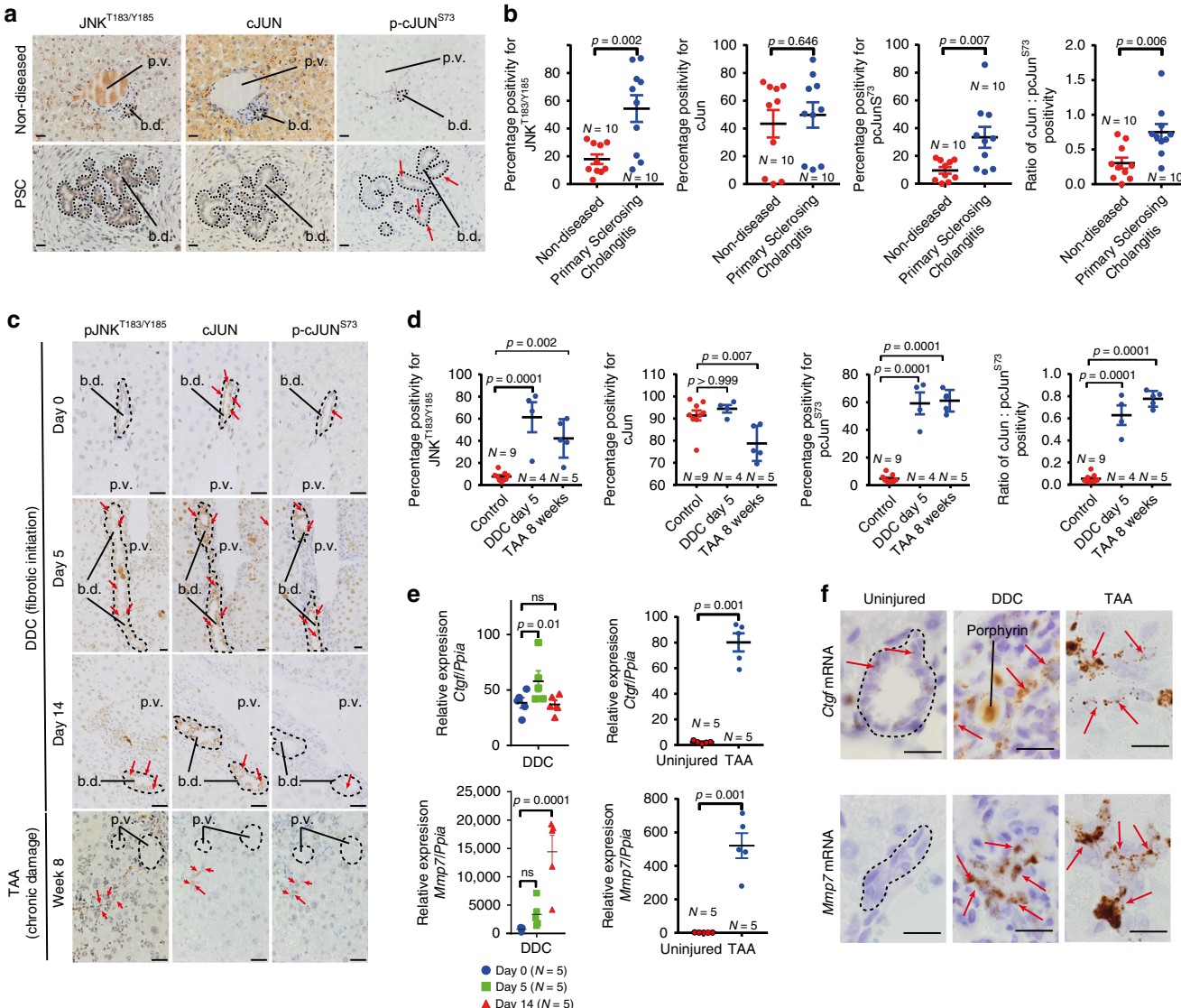

**Fig. 1 Activation of the Jnk–Jun signal in biliary disease. a** Immunohistochemistry on serial sections of either non-diseased (upper panels) or primary sclerosing cholangitis tissue (bottom panels) stained for phosphorylated JNK[T183/Y185], total c-JUN and phosphorylated c-JUN[S73]. Dotted lines demarcate the boundary of bile ducts. Red arrows identify biliary epithelial cells with positive phosphorylated c-JUN[S73] expression. **b** Quantification of phosphorylated JNK[T183/Y185], total c-JUN or phosphorylated c-JUN[S73] and the quantification of c-JUN:phospho-c-JUN[S73] in normal human and primary sclerosing cholangitis tissue. **c** Immunohistochemistry of phosphorylated JNK[T183/Y185], total c-JUN and phosphorylated c-JUN[S73] in a time course of DDC injury to model fibrotic initiation or in established fibrosis induced by treatment with TAA. Red arrows denote positive nuclei. Dotted lines demarcate biliary epithelial cells and bile ducts. **d** Quantification of phosphorylated JNK[T183/185], total c-JUN, phosphorylated c-JUN[S73] and the ratio of total c-JUN: phosphorylated c-JUN[S73] in the DDC and TAA models. **e** mRNA expression of Wnt-PCP target genes in mice undergoing DDC and TAA-induced biliary injury, normalised to the housekeeping gene, *Ppia*. **f** RNAScope of mRNA for *Ctgf* and *Mmp7* in healthy bile ducts and following bile duct injury. Red arrows denote RNAScope positivity. Scale bar = 50 μm. p.v.—portal vein, b.d.—bile duct. Porphyrin (labelled) accumulation in the duct is a consequence of the DDC model. Source data are provided as a Source Data file. In graphs where two groups are included, a Student's *t* test is used. When comparing multiple groups, a one-way ANOVA with post hoc correction for multiple testing is used. In dot plots, data are presented as mean ± S.E.M. Each data point (N) represents an individual animal or patient.

including *Axin2* were not changed following Porcupine inhibition (Supplementary Fig. 3a). There was an obvious reduction in glutamine synthetase (GS) positivity, a known target of Wnt-β-catenin signalling in hepatocytes (Supplementary Fig. 3b), confirming that Wnt-ligand inhibition across the liver had been successful. In proliferating BECs (in both the TAA and DDC models) Porcupine-i results in a significant reduction in phospho-JNK[T183/Y185] and phospho-c-JUN[S73] positivity, indicating that in biliary epithelial cells the activation of Jnk/c-Jun signalling is, in part, regulated by Wnt ligands (Fig. 2c).

Having defined that Porcupine-i alters Jnk/c-Jun signalling in BECs we sought to then determine whether the inhibition of Wnt-ligand secretion affects the regeneration of the biliary tree. Whilst the number of BECs (stained for Keratin-19) and fibroblasts (stained for Desmin) remained the same between Porcupine-i and control mice (Fig. 2d), the amount of total fibrillar collagen (determined through staining for Picrosirius Red) and specifically Collagen-1 that is deposited during bile duct regeneration is significantly reduced following Porcupine inhibition (Fig. 2d).

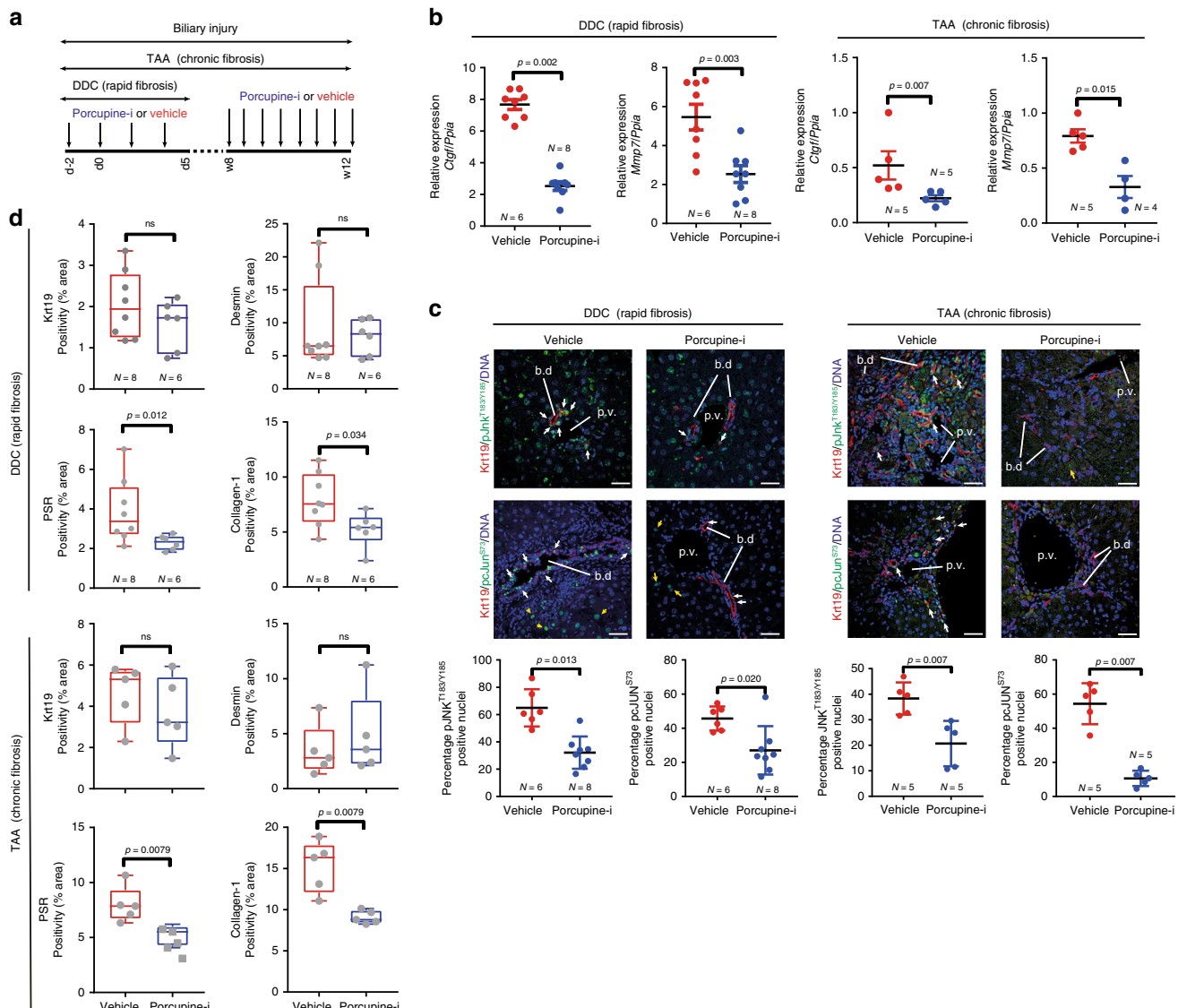

**Fig. 2 Therapeutic inhibition of Wnt reduces scarring. a** Schematic of Porcupine inhibition with Porcupine-i during bile duct regeneration induced by DDC or TAA. **b** mRNA expression of Wnt-PCP pathway target genes *Ctgf* and *Mmp7* in isolated bile ducts following Porcupine-i treatment, normalised to the housekeeping gene *Ppia*. **c** Immunofluorescent staining of biliary epithelial cells (Keratin-19-positive, red) with phosphorylated JNK$^{T183/Y185}$ (upper panels) or phosphorylated c-JUN$^{S73}$ (lower panels), green. White arrows denote positivity in biliary epithelial cells, yellow arrows show positivity in other, non-keratin-19-positive cells. Positivity of phosphorylated JNK$^{T183/Y185}$ and phosphorylated c-JUN$^{S73}$ quantified in dot plots and represented as a proportion of total biliary epithelial cells counted. **d** Histological quantification of Keratin-19 (biliary epithelial cells), Desmin (fibroblasts), Picrosirius Red, PSR (fibrilar collagens) and Collagen-1 in livers of mice treated with either DDC or TAA, following Porcupine inhibition. Scale bar = 50 μm. p.v.—portal vein, b.d.—bile duct. Source data are provided as a Source Data file. In all comparisons, a Student's *t* test is used. Box–whisker plots represent min–max range of the data. In dot plots, data are presented as mean ± S.E.M. Each data point (N) represents an individual animal.

Given that the inhibition of Wnt-ligand secretion is sufficient to alter Jnk/c-Jun signalling and regulate biliary fibrosis, we then sought to define the origin of the Wnt ligands involved in this process. Previous work from our lab and others has shown that myeloid cells (including macrophages) are a source of multiple Wnt ligands[3,9,45], including the archetypal Wnt-PCP ligand, Wnt5a[46], and indeed, in CD45$^+$/CD11b$^+$ macrophages isolated from the livers of mice with biliary injury, we found that macrophages expressed WNT5A. Moreover, in macrophages from mice treated with porcupine-i, 50% of cells retained WNT5A in their ER compared with 32% in vehicle-treated animals (Supplementary Fig. 3c-f). *WNT5A* is transcriptionally upregulated in liver patients with PSC compared with livers from patients without underlying disease (Supplementary Fig. 4a), and

similarly, in mouse models of biliary regeneration *Wnt5a* transcript expression is induced following injury (Fig. 3a). In these murine models of bile duct regeneration and BEC proliferation, CD68-positive macrophages, but not Desmin-positive myofibroblasts, in close proximity to BECs express *Wnt5a* mRNA (Supplementary Fig. 4c) and WNT5A protein[47] (Fig. 3b). Moreover, in PSC patient tissues WNT5A-positive cells localise to the scars surrounding the bile duct (Supplementary Fig. 4b), suggesting that the provision of Wnt ligand by macrophages could promote biliary fibrosis.

To understand whether Wnt5a-expressing myeloid cells regulate biliary scarring, we made use of a mouse line in which Cre recombinase is expressed by myeloid lineages to delete Wnt5a specifically in these cells (LysMCre::*Wnt5a*$^{flox}$,

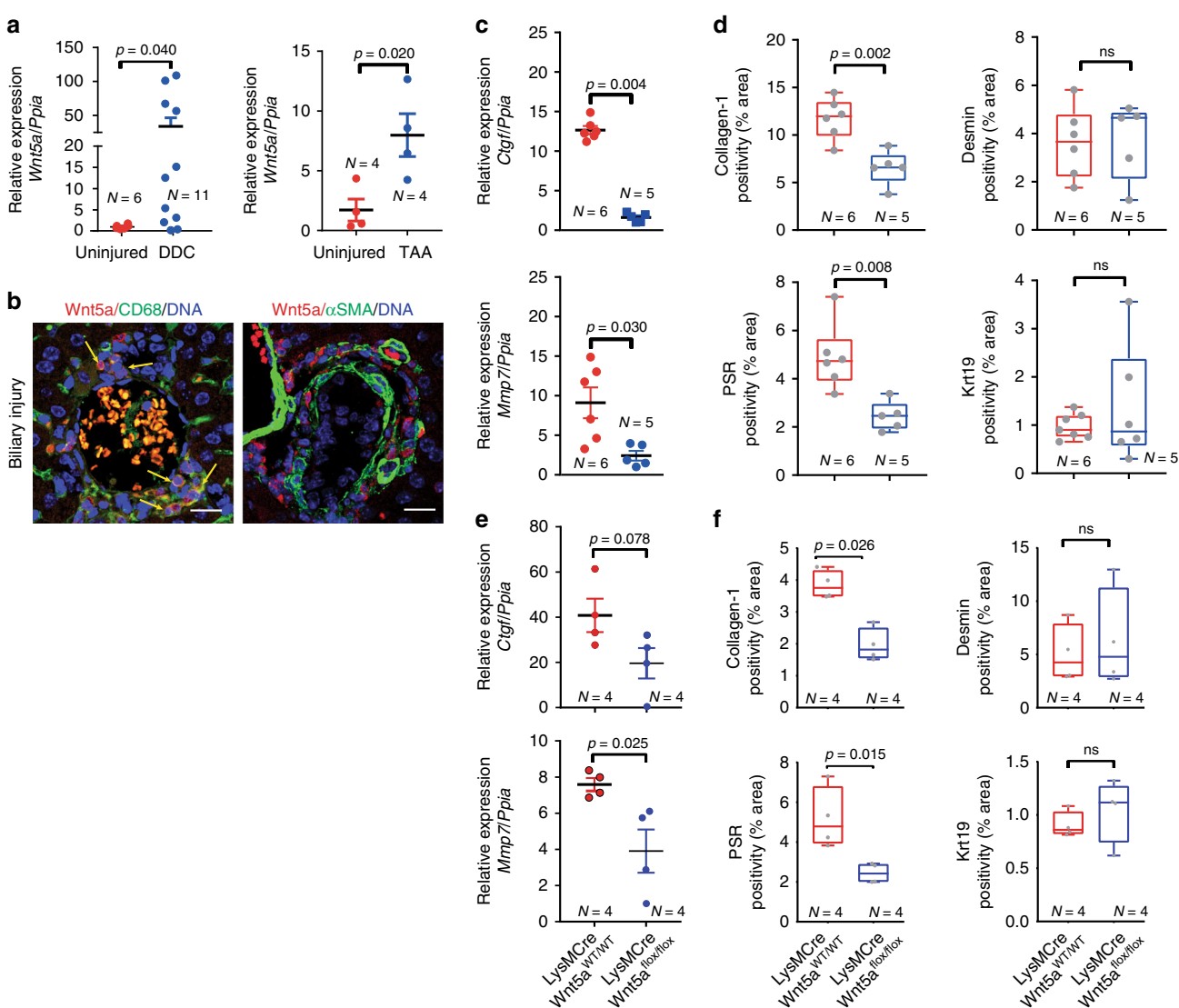

**Fig. 3 Myeloid Wnt5a promotes biliary scar deposition. a** mRNA expression of Wnt5a in whole liver from mice with either DDC (left panel) or TAA (right panel) induced biliary disease. **b** Immunofluorescence showing the localisation of WNT5A protein (red) to CD68-positive cells (green, left panel), not αSMA-positive cells (green, right panel), yellow arrows show dual-positive cells. **c** mRNA from DDC-treated mice showing loss of Wnt-PCP target genes, Ctgf and Mmp7, normalised to Ppia in isolated bile ducts following deletion of Wnt5a from monocyte lineages. **d** Histological quantification of Keratin-19 (biliary epithelial cells), Desmin (fibroblasts), Picrosirius Red, PSR (fibrillar collagens) and Collagen-1 in livers of mice treated with DDC following myeloid-specific deletion of Wnt5a. **e** mRNA from TAA-treated mice showing loss of Wnt-PCP target genes, Ctgf and Mmp7 normalised to Ppia in isolated bile ducts following deletion of Wnt5a from monocyte lineages. **f** Histological quantification of Keratin-19 (biliary epithelial cells), Desmin (fibroblasts), Picrosirius Red, PSR (fibrillar collagens) and Collagen-1 in livers of mice treated with TAA following myeloid-specific deletion of Wnt5a. Scale bar = 50 μm. Source data are provided as a Source Data file. In graphs where two groups are included, a Student's t test is used. Box–whisker plots represent min–max range of the data. In dot plots, data are presented as mean ± S.E.M. Each data point (N) represents an individual animal.

Supplementary Fig. 4d–g). Following biliary injury with either DDC or TAA, isolated bile ducts showed a reduced mRNA expression of Ctgf and Mmp7 when myeloid Wnt5a was deleted compared with ducts isolated from mice in which Wnt5a was intact (Fig. 3c, e), but did not show changes in Wnt-β-catenin pathway target expression (Supplementary Fig. 4h). Moreover, in both models of bile duct regeneration, the amount of PSR staining and Collagen-1 immunostaining was significantly reduced following Wnt5a deletion in myeloid cells, without affecting the number of keratin-19-positive BECs or Desmin-positive myofibroblasts (Fig. 3d, f). Interestingly, in our acute (DDC) model of biliary regeneration, high Wnt5a expression does not persist throughout injury. When Wnt5a is deleted from myeloid cells specifically at later time points in this model, the changes in Collagen-1 and PSR

are lost, suggesting that either Wnt5a is only required for the initiation of ductular fibrosis in this context or that other Wnt ligands or signalling pathways compensate for Wnt5a loss in myeloid cells.

**Wnt-PCP receptors regulate ductular scarring**. There is a high level of redundancy within the Wnt-ligand family[48] and conflicting data as to whether Wnt ligands are faithful activators of particular downstream receptors. Therefore, we sought to validate our findings that Wnt ligands and particularly Wnt5a regulate biliary scarring through Wnt-PCP activation. In Wnt-PCP signalling a number of receptors, including Frizzled receptors[49] and orphan receptor tyrosine kinases such as ROR1/2 and PTK7 are

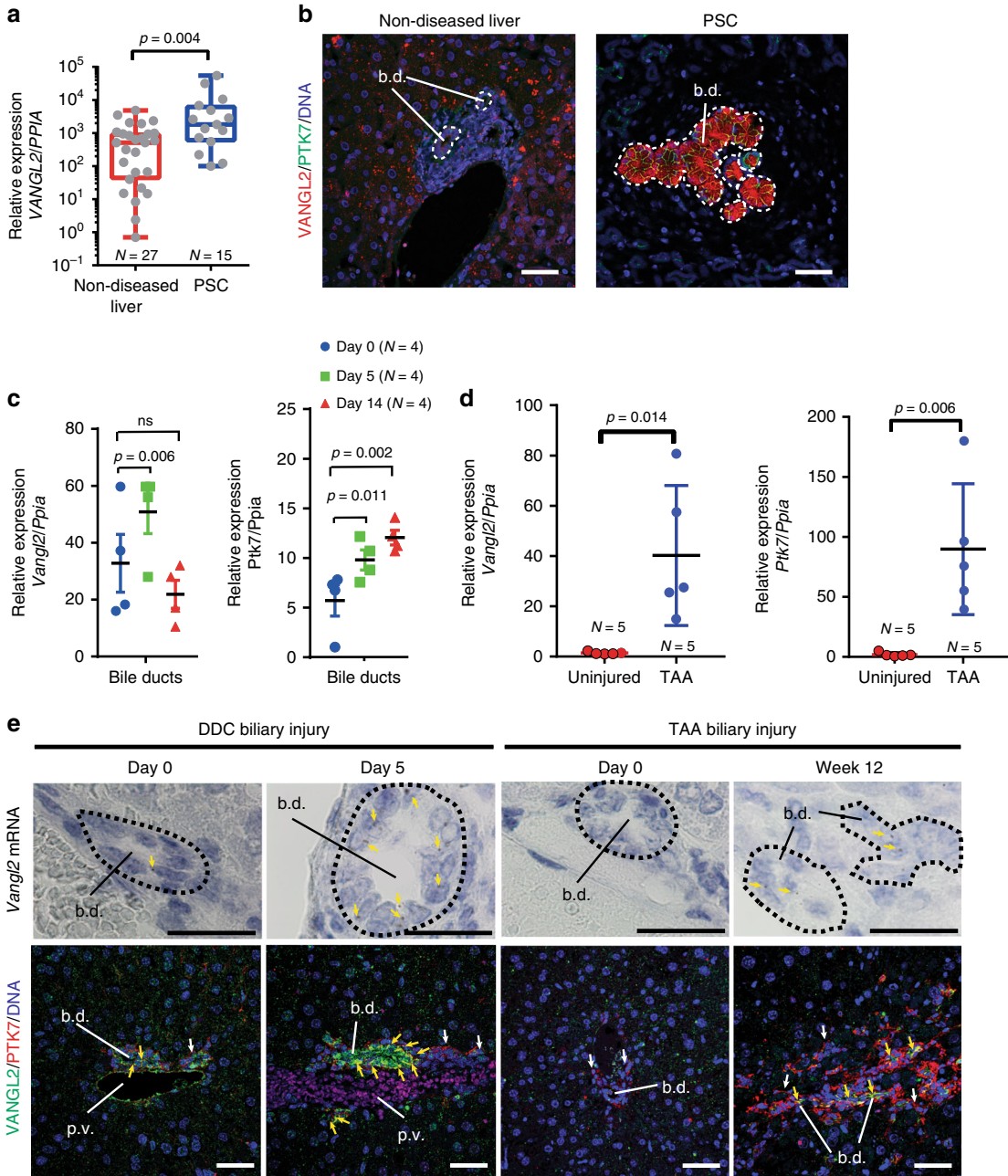

**Fig. 4 Wnt-PCP receptors are expressed by regenerating bile ducts. a** mRNA expression of *VANGL2* in healthy liver tissue and tissue from patients with primary sclerosing cholangitis. **b** Immunofluorescence of VANGL2 (red) and PTK7 (green) in tissue from healthy liver and patients with primary sclerosing cholangitis. Dotted lines denote bile ducts. **c** mRNA expression of *Vangl2* and *Ptk7* normalised to *Ppia* in isolated bile ducts and whole liver from DDC-induced ductular regeneration. **d** mRNA expression of *Vangl2* and *Ptk7* in animals with TAA-induced biliary injury normalised to *Ppia*. **e** Upper panels: RNAScope of *Vangl2* mRNA in DDC and TAA-induced bile duct regeneration (yellow arrows denote positivity and dotted lines denote bile ducts). Lower panels: immunohistochemistry for VANGL2 (green) and PTK7 (red) in healthy bile ducts, or bile ducts injured with either DDC or TAA. White arrows denote PTK7 single positive stromal cells. Yellow arrows denote biliary epithelial cells that express both VANGL2 and PTK7 in biliary epithelial cells. Scale bar = 50 μm. Source data are provided as a Source Data file. In graphs where two groups are included, a Student's *t* test is used. When comparing multiple groups, a one-way ANOVA with post hoc correction for multiple testing is used. Box–whisker plots represent min–max range of the data. In dot plots, data are presented as mean ± S.E.M. Each data point (N) represents an individual animal or patient.

known to bind Wnt ligands[50]. These then converge on the scaffolding proteins VANGL1 and VANGL2, which become phosphorylated[51], a process that is necessary for Wnt-PCP activity. *VANGL1* and *VANGL2* are transcriptionally upregulated in PSC patients (Fig. 4a and Supplementary Fig. 5a). Moreover, in PSC patients, VANGL2 and PTK7 proteins localise to the plasma membranes of proliferating BECs (Fig. 4b), but not in healthy bile ducts suggesting that upregulation of Wnt-PCP pathway components occurs in biliary disease.

As Vangl2 is functionally dominant over its homologue Vangl1[52], we elected to study whether Wnt signalling via Vangl2 regulates the formation of biliary scars. Similar to human, healthy mouse bile ducts express low levels of *Vangl1* and *Vangl2* at the transcriptional level, and positive staining for VANGL1/2

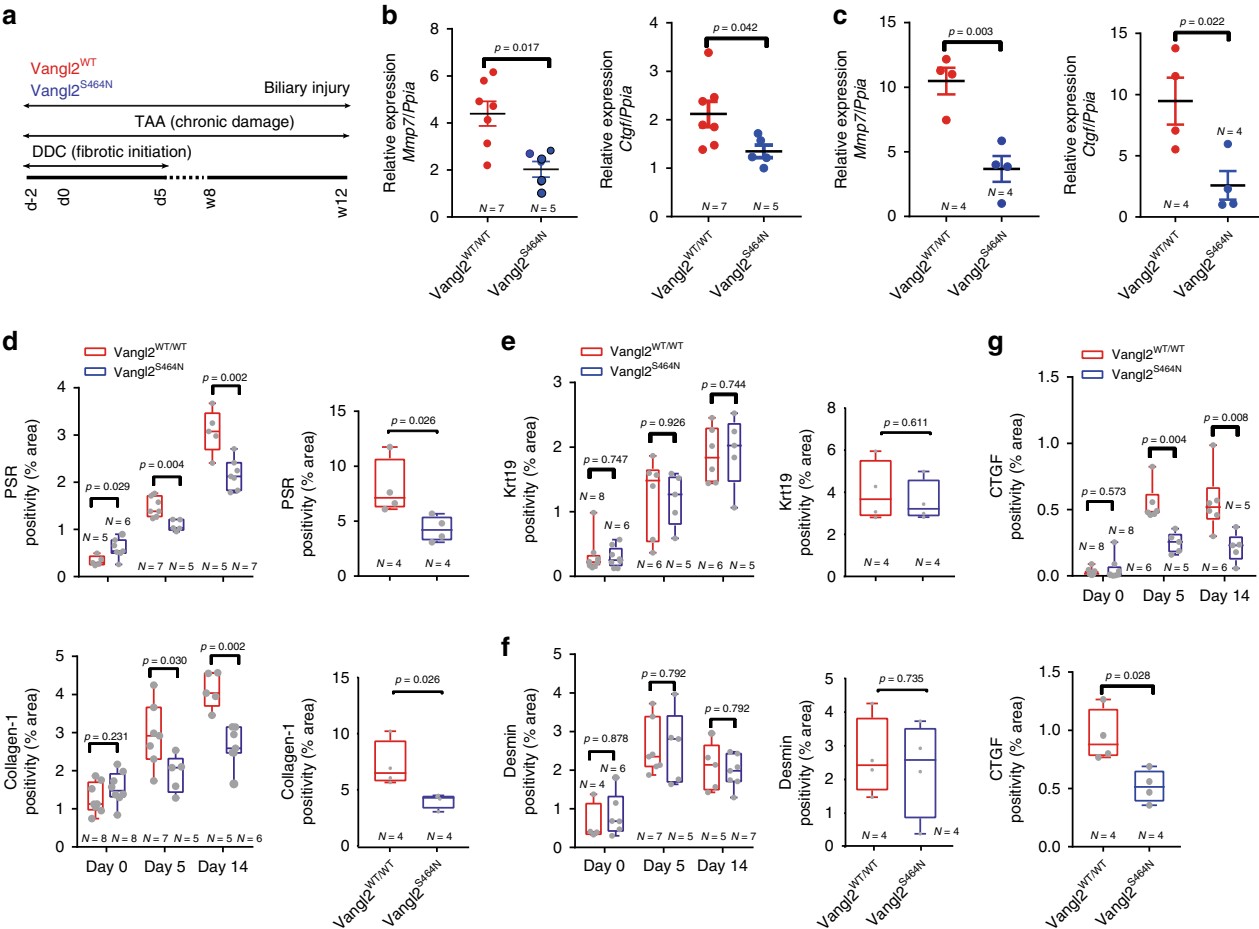

**Fig. 5 Inhibition of Wnt-PCP reduces the formation of biliary scars. a** Schematic representing the treatment strategy of Vangl2$^{S464N}$ mutant mice treated with either DDC or TAA to initiate bile duct damage and regeneration. **b** Relative mRNA expression of Wnt-PCP target genes *Ctgf* and *Mmp7* in DDC and **c** TAA-induced biliary regeneration normalised to *Ppia*. **d** Image quantification of Picrosirius Red, PSR (fibrillar collagen) and Collagen-1 in DDC or TAA-induced biliary injury. **e** Quantification of the number of BECs (through Keratin-19 immunohistochemistry) following DDC (left panel) or TAA (right panel) in Vangl2$^{S464N}$ mutant mice compared with controls. **f** Desmin positivity (fibroblasts) in mice with DDC or TAA-induced biliary regeneration. **g** Image analysis and quantification of CTGF protein levels in mice with DDC (upper panel) or TAA- (bottom panel) induced biliary injury. Source data are provided as a Source Data file. In graphs where two groups are included, a Student's *t* test is used. When comparing multiple groups, a two-way ANOVA with post hoc correction for multiple testing is used. Box–whisker plots represent min–max range of the data. In dot plots, data are presented as mean ± S.E.M. Each data point (N) represents an individual animal.

proteins in the plasma membranes of BECs is infrequent (Fig. 4c–e and Supplementary Fig. 5b). Similarly, in healthy BECs, *Ptk7* expression is low, although PTK7-positive portal fibroblasts can be seen surrounding healthy ducts (Fig. 4e). Following bile duct injury with either DDC or TAA, *Vangl2* and *Ptk7* mRNA levels increase in isolated bile ducts (Fig. 4e and Supplementary Fig. 5b), and VANGL2 and PTK7 proteins co-localise on the plasma membrane of BECs specifically (Fig. 4e, lower panels).

Wnt-PCP signalling can be effectively inhibited through modulating Vangl2 activity using a semi-dominant hypomorphic Vangl2 mutant, *Vangl2$^{S464N}$*, also known as Looptail[22]. Mice homozygous for this mutation die shortly after birth; however, heterozygous mice (*Vangl2$^{S464N/+}$*, hereafter denoted as *Vangl2$^{S464N}$*) are viable, reach adulthood and have been used to determine the role of Vangl2 in a number of adult contexts[53]. In isolated bile ducts from *Vangl2$^{S464N}$* mice given either DDC or TAA injury (Fig. 5a), we found a significant reduction in the mRNA expression of *Mmp7* and *Ctgf* (Fig. 5b, c), indicating that in both short-term and long-term fibrotic injury, signalling downstream of BEC-expressed VANGL2 regulates *Ctgf* and

*Mmp7* levels. Having shown that mutations in *Vangl2* phenocopy the transcriptional changes seen with Porcupine inhibition (Fig. 2) we sought to define whether mice carrying the *Vangl2$^{S464N}$* mutation had a reduced ability to form bile duct scars following injury. Following DDC or TAA injury, mice with wild-type *Vangl2* developed progressive ductular fibrosis. In littermates that carried a heterozygous mutation, however, the level of scarring was significantly reduced (Fig. 5d). Interestingly, mutations in *Vangl2* did not affect the number of BECs (Fig. 5e), nor did reduced VANGL2 function result in changes in the number of Desmin-positive fibroblasts (Fig. 5f) found within the liver. In concordance with the mRNA expression (Fig. 5b, c), the levels of CTGF protein were significantly reduced in *Vangl2$^{S464N}$* mice following bile duct injury compared with control littermates (Fig. 5g). Given that the *Vangl2$^{S464N}$* mutant mouse phenocopies Wnt-ligand loss following Porcupine inhibition, we suggest that Wnt-PCP signalling is required for the establishment of biliary scars in the mouse.

The *Vangl2$^{S464N}$* mutation is constitutive, and therefore we could not preclude that there were developmental deficiencies in bile duct development and patterning, which are simply

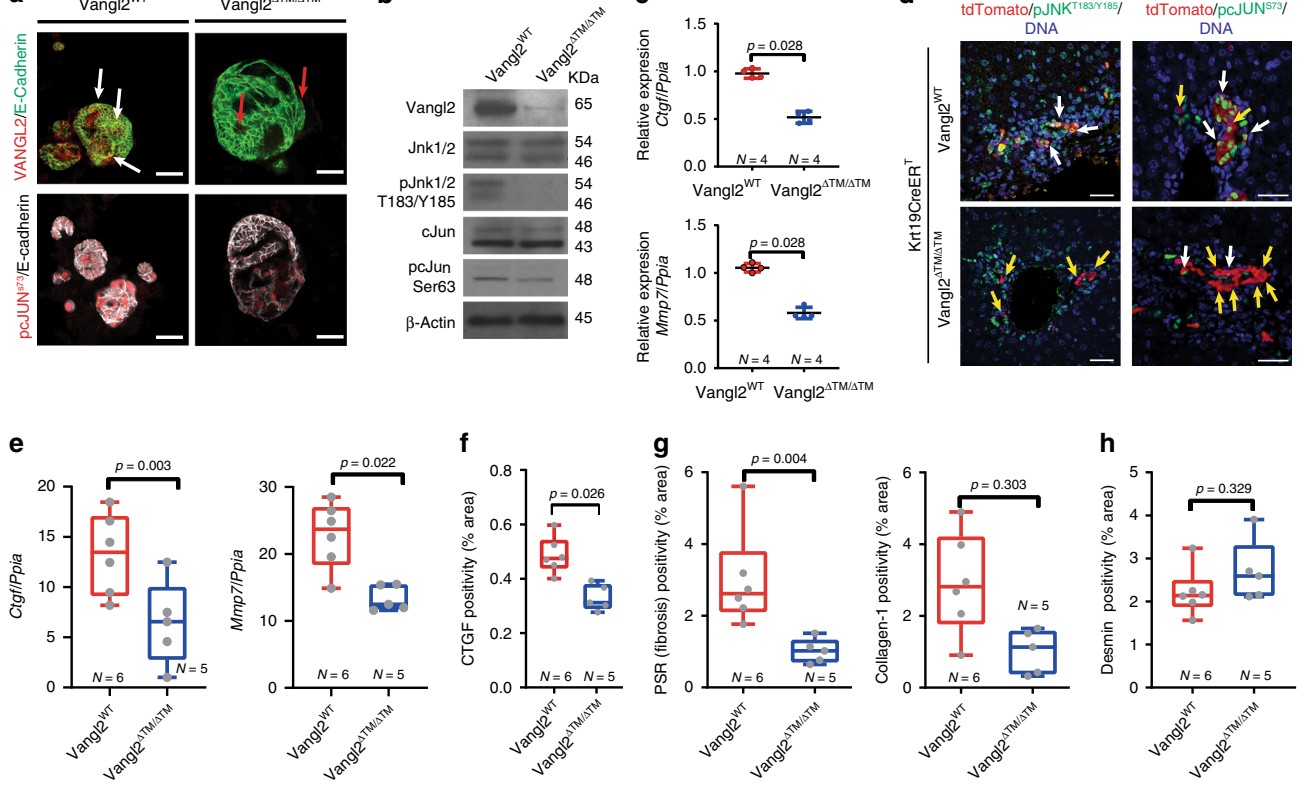

**Fig. 6 Wnt-PCP loss alters Jnk-c-Jun signalling in biliary cells. a** Biliary organoids from Vangl2$^{WT}$ mice or Vangl2$^{flox}$ mice (following treatment with Cre lentivirus, known as Vangl2$^{\Delta TM}$), immunostained for VANGL2 (red) membrane-bound E-cadherin (green), upper panels. White arrows denote co-immunofluorescence, whereas red arrows denote residual small Vangl2 puncta, not localised to the cell membrane. Lower panels represent the same conditions as described in (**a**), but stained for phosphorylated c-JUN$^{S73}$ (red) and E-cadherin, white. **b** Western blot of proteins from Vangl2$^{WT}$ and Vangl2$^{\Delta TM}$ biliary organoids. **c** Relative mRNA expression Wnt-PCP target genes, *Ctgf* and *Mmp7* normalised to the housekeeping gene *Ppia* from Vangl2$^{WT}$ and Vangl2$^{\Delta TM}$ biliary organoids. **d** Immunofluorescent staining of tdTomato (red), phosphorylated Jnk$^{T183/Y185}$ or phosphorylated c-JUN$^{S73}$ (green) and DNA (blue) in mice where Vangl2 has been deleted specifically in biliary epithelial cell (Keratin-19-CreER$^{T}$::Vangl2$^{flox}$, following Cre activation known as Keratin-19-CreER$^{T}$::Vangl2$^{\Delta TM}$). White arrows denote dual-positive cells and yellow arrows denote cells that are only tdTomato (red) positive. **e** mRNA expression of Ctgf and Mmp7 normalised to the housekeeping gene *Ppia* in bile ducts isolated from Keratin-19-CreER$^{T}$::Vangl2$^{\Delta TM}$ mice following DDC injury. **f** Histological quantification of CTGF protein in Keratin-19-CreER$^{T}$::Vangl2$^{\Delta TM}$ mice following DDC-induced bile duct injury. **g** Histological quantification of total fibrillar collagen (Picrosirius Red, PSR), Collagen-1 and **h** Desmin (fibroblasts) following DDC treatment of Keratin-19-CreER$^{T}$:: Vangl2$^{\Delta TM}$ mice. Scale bar = 50 μm. Source data are provided as a Source Data file. In all graphs, a Student's *t* test is used. Box–whisker plots represent min–max range of the data. In dot plots, data are presented as mean ± S.E.M. Each data point (N) represents an individual animal. In (**c**), each N represents an experimental replicate.

exacerbated during ductular regeneration. To overcome this, we utilised a transgenic mouse line in which Vangl2 can be specifically deleted in cells (Vangl2$^{flox}$)[54].

To confirm that *Vangl2* can be deleted in the BEC lineage, we isolated bile ducts from *Vangl2$^{flox/flox}$* mice or wild-type littermates and grew these as organoids (Fig. 6a). Following infection with lentiviral-Cre (Lv-Cre), BECs in these organoids recombine floxed alleles in a mosaic way (Supplementary Fig. 6a). However, recombined cells can be enriched and produce relatively pure organoid cultures. BEC organoids derived from *Vangl2$^{flox/flox}$* mice grow normally following expression of Cre, however they have significantly reduced *Vangl2* mRNA expression (Supplementary Fig. 6b). Furthermore, VANGL2 protein levels are reduced and VANGL2 no longer localises to the plasma membrane of BECs in these organoids (Fig. 6a, b, from herein these organoids are known as Vangl2$^{\Delta TM}$). Following *Vangl2* loss, Vangl2$^{\Delta TM}$ organoids show reduced levels of nuclear phospho-c-JUN$^{S73}$ (Fig. 6a, lower panels), and whilst the levels of total JNK and c-JUN remain unchanged in *Vangl2$^{WT}$* versus *Vangl2$^{\Delta TM}$* organoids, BECs in organoids lacking *Vangl2* have

notably reduced levels of phosphorylated JNK$^{T183/Y185}$ and have a 40% reduction of phosphorylated c-JUN$^{S73}$ (Fig. 6b). Furthermore, VANGL2 loss significantly reduces the mRNA levels of *Mmp7* and *Ctgf* (Fig. 6c), confirming that the levels of these transcripts relate to signalling through Vangl2.

Having confirmed that Vangl2-specific deletion in BECs alters Wnt-PCP signalling in vitro, we generated a mouse in which Vangl2 could be specifically deleted in BECs following administration of tamoxifen (K-19CreER$^{T}$::Vangl2$^{flox}$). In K-19CreER$^{T}$:: Vangl2$^{flox}$ mice, following injury, we failed to detect phospho-JNK$^{T183/Y185}$ or phospho-c-JUN$^{S73}$ in Vangl2$^{\Delta TM}$ BECs (Fig. 6d). Importantly, Vangl2$^{\Delta TM}$ cells (lineage traced with RFP) are retained in vivo in the bile duct following injury (Supplementary Fig. 6c and d), indicating that *Vangl2* is not an essential gene for BEC survival. Yet, following *Vangl2* deletion, the amount of *Mmp7* and *Ctgf* mRNA in isolated bile ducts and CTGF protein was significantly reduced (Fig. 6e, f). Moreover, the levels of total fibrillar collagen assessed through Picrosirius Red staining and Collagen-1 specifically, surrounding these Vangl2$^{\Delta TM}$ BECs, were also reduced (Fig. 6g) without affecting the number of portal

fibroblasts (Fig. 6h), confirming that lineage-specific deletion of *Vangl2* in BECs is sufficient to reduce biliary scarring and validating the phenotype found using *Vangl2$^{S464N}$* mutant mice.

## Discussion

Canonical Wnt-β-catenin signalling is widely regarded as a principal regulator of the cell cycle in adult tissue homoeostasis[55] and regeneration[35], where the stabilisation of β-catenin and its translocation into the nucleus promotes the transcription of a number of pro-proliferative and pro-survival pathways in a plethora of tissues. In the context of cancers too, the role of Wnt-β-catenin has been widely and extensively studied[45,56] as mutations in β-catenin and in its hallmark regulators APC and Axin promote deregulated cell proliferation and are now considered oncogenes in a wide array of contexts[1,57,58].

Wnt ligands do however, act through a number of highly conserved developmental pathways[59], which likely evolved to regulate the formation of the anterior–posterior axis in the earliest multicellular organisms, and establish polarity in groups of cells[60,61]. This pathway, the Wnt-Planar Cell Polarity (Wnt-PCP) pathway, is necessary for establishing any number of axes within the embryo, and through its control of cell shape, migration and planar cell movement is absolutely required for normal organogenesis[38,62,63]. The role of Wnt ligands in establishing these body plans likely predates their role in activating β-catenin-dependent transcription, yet whether Wnt-PCP signalling plays a role in adult homoeostasis and regeneration remains largely mysterious, and work unpicking the role of Wnt-PCP in adult disease and tissue regeneration remains in its infancy.

The bile duct, like other epithelial structures in the liver, has a remarkable regenerative capability, and following injury, biliary epithelial cells (BECs) begin to proliferate and form new tubular networks, ultimately restoring flow though the duct and maintaining the function of the liver[27]. The role of Wnt signalling in the context of BEC proliferation and bile duct regeneration has been contested, with a number of early papers demonstrating activation of the canonical, Wnt-β-catenin-dependent pathway in these cells[11,28]. Whether Wnt-β-catenin is as important as these early papers suggested remains contentious, as we and others have failed to find nuclear translocation of β-catenin or activation of canonical Wnt-β-catenin receptors in BECs during bile duct regeneration[3,35]. Furthermore, recent studies using single-cell RNA (scRNA) sequencing have failed to find an obvious canonical Wnt signature in BECs[14,15], and deletion of the Wnt-potentiating R-Spondin (RSPO)-LGR system in BECs had little effect on their proliferation[14]. These data are not unequivocal, as recent work has indicated that RSPO can function independently of LGR proteins[64]. Though if this is the case in BECs they would stand out as unusual as the majority of Wnt-responsive cells in the adult have been shown to require the RSPO-LGR rheostat[35,55,65]. Several papers have now suggested that Notch and Yap signalling, not canonical Wnt signalling, are the favoured routes to promote BEC proliferation[3,14,15,66]. Our data presented here are in agreement with this observation; however, whereas other groups fail to see *Axin2* transcripts in BECs[14,15], we find infrequent transcripts that do not increase with injury, and therefore we conclude that canonical Wnt-β-catenin signalling is not upregulated in patients with biliary disease or animal models.

There is little doubt that Wnt ligands are expressed during biliary regeneration[3,13,15,28], and from our data in this paper and those data from other groups it appears that in the context of bile duct injury and repair at least that myeloid-derived Wnts play an important role in regulating this process[42,43]. Importantly, others have shown that BEC proliferation is dependent on Wnt ligands, but that this proliferation in BECs is β-catenin independent,

suggesting that non-canonical Wnt signalling pathways, including but not limited to Wnt-PCP, regulate cholangiocyte proliferation[13]. In this study, we repeatedly fail to see a reduction in cholangiocyte number when we inhibit Wnt-PCP signalling, as would be expected based on previous data[13]. We cannot exclude, however, the possibility that there are compensatory proproliferative mechanisms that take place following pharmacological inhibition of Wnt-ligand production, nor can we exclude the possibility that BEC proliferation is driven through a noncanonical or atypical Wnt pathway that does not depend on Vangl2, such as Wnt-STOP or Wnt-YAP/TAZ signalling[67,68].

In this study, we do, however, find a consistent and reproducible suppression of bile duct fibrosis when Wnt-PCP signalling is inhibited in BECs. In this model, the binding of Wnt ligands, through Wnt-PCP receptor activation, including Ptk7 and Vangl2, results in the phosphorylation of Jnk, which promotes c-Jun-dependent transcription. These observations are concordant with scRNA data in which the AP-1 transcription factor network is enriched in BECs[15]. Inhibition of Wnt ligands, either genetically or using a therapeutic Wnt inhibitor, significantly decreases the amount of biliary scarring that occurs following injury, leading us to demonstrate that Wnt-PCP signalling is upstream of fibrogenic cytokine production. Surprisingly, heterozygous loss of *Vangl2*, through the *Vangl2$^{S464N}$* mutation, is sufficient to reduce biliary scarring, suggesting that this system requires tight regulation. Emerging data from other chronically diseased tissues indicate that Wnt-PCP signalling can play a role in the establishment and maintenance of scarring and deposition of the extracellular matrix[20,53,69,70]; here we show that Wnt-PCP is also involved in the formation of biliary scars in the adult liver.

Adult tissue repair, by its very nature, requires fine balance between epithelial regeneration and scarring. While fibroblasts in the regenerative microenvironment are the primary cells that make scar tissue, we and others have previously shown that the fibroblast microenvironment in biliary regeneration provides a number of signals that BECs require to maintain both their proliferation and lineage specification[3,4,66]. Deleting these profibrogenic cells or inhibiting their activation/proliferation would therefore have significant consequences for the regenerating biliary epithelium and could ultimately prevent ductular regrowth following injury. Inhibition of Wnt signalling or loss of Wnt-PCP receptors on proliferating BECs does not affect the formation of the cellular regenerative microenvironment, with pro-regenerative fibroblast numbers and the numbers of BECs remaining constant following inhibition. Our data demonstrate that it is possible to pharmacologically uncouple regeneration from scar fomation in the diseased bile duct and suggest that inhibition of Wnt-ligand production should be considered as a therapy in chronic biliary diseases.

## Methods

**Mouse models**. All mice were maintained in 12-h light/dark cycles and had access to food and water ad libitum in accordance with UK Home Office Regulations. All experiments were performed under UK Home Office licence PPL: 70/8150 held by Dr Luke Boulter; mice were euthanised by an escalating dose of $CO_2$. Animal experiments were approved by the University of Edinburgh Animal Welfare Ethical Review Board (AWERB).

For the DDC mouse model, 6–8-week-old, male CD1 mice were fed 0.1% 5-diethoxycarbonyl-1,4-dihydrocollidine (DDC) in their diet for up to 14 days and provided with normal drinking water. Mice that drop >20% of their bodyweight were given DDC food softened with water. For the TAA model, mice were given 400 mg/L thioacetamide in sweetened water for up to 12 weeks. Thioacetamide water was present throughout the model and mice would access this ad libitum. In studies where mice received the Porcupine inhibitor, LGK974 experimental mice were dosed with 5 mg/kg LGK974, twice daily via oral gavage (the vehicle for this was 0.5% methylcellulose and 0.5% Tween-80 in $dH_2O$). Mice were dosed for 48 h prior to the administration of DDC to ensure inhibited Wnt-ligand production prior to injury. Effectiveness of LGK974 was confirmed through reduction of GS

staining adjacent to central veins and is known to be Wnt-ligand dependent. Finally, to generate livers that overexpressed β-catenin[NF90], we used a hydrodynamic model in which 20 μg of pT3-EF1aH N90-beta-catenin (a gift from Xin Chen (Addgene plasmid # 86499; http://n2t.net/addgene:86499; RRID: Addgene_86499)) was co-injected with 6 μg of PT2/C-Luc//PGK-SB13 (a gift from John Ohlfest (Addgene plasmid # 20207; http://n2t.net/addgene:20207; RRID: Addgene_20207)). These plasmids were delivered in 10% w/v physiological saline in <7 s.

Vangl2[eGFP] mice were provided by Dr Ping Chen, Emory University, USA, and were maintained on a CD1 background. Looptail mice (Vangl2[S464N], Harwell, UK and provided by Dr Charlotte Dean) were maintained on a C3H background, and throughout these studies, wild-type littermates served as a control. Krt19CreER[T]:R26RtdTomato:Vangl2[flox/flox] mice were generated by crossing the Krt19CreER[T] mouse line initially with the tdTomato Ai14 line from Jax labs. Mice heterozygous for the K-19CreER[T] knock-in and homozygous for tdTomato were then crossed with Vangl2flox mice, which were generated and kindly provided by Prof Deborah Henderson, Newcastle. In this study, all Krt19CreER[T]:R26RtdTomato:Vangl2[flox/flox] mice were positive for tdTomato and CreER[T] and were then homozygous, heterozygous or wild type for the Vangl2[flox] allele. Induction of Cre was achieved through repeat administration of tamoxifen (three times for 1 week) at a concentration of 4 mg per mouse for each injection. Tamoxifen was diluted in 10% molecular-grade ethanol: 90% corn oil. LysMCre:Wnt5a[flox/flox] mice were generated by crossing the LysMCre line kindly provided by Dr Steven Jenkins, University of Edinburgh, with Wnt5a[flox/flox] mice purchased from Jax labs. Mice were maintained as Cre positive and homozygous, heterozygous or wild type for the floxed allele. All genetically engineered mice were genotyped by Transnetyx Inc.

**Human tissue**. Five-micrometre unstained sections of formalin-fixed paraffin-embedded explant fibrotic human liver from the deep right lobe of native explants from patients with primary sclerosing cholangitis undergoing allograft liver transplantation were received. Explant livers were received fresh on ice and fixed immediately by the installation of 4% v/v neutral buffered formalin (Genta Medical, UK) via the hepatic veins as part of the standard specimen pathway. Human tissue was obtained by approved application to the Lothian NRS Human Annotated Bioresource that is authorised to provide unconsented anonymised tissue under ethical approval number 15/ES/0094 from the East of Scotland Research Ethics Service REC 1. Informed consent was waived due to the historical nature of these samples.

RNA isolation and RT-PCR: RNA was isolated from both tissue and organoids by using a modified Trizol Method. Briefly, tissue was homogenized in Trizol and RNA collected from the aqueous phase using chloroform extraction. RNA was precipitated from the aqueous phase by using isopropanol. Precipitated RNA was applied to an RNeasy Mini Spin Column (Qiagen) and the manufacturer's protocol was followed. RNA was suspended in RNAse/DNAse-free $H_2O$ and quantified using a Nanodrop (Thermofisher). In all, 1 μg of total RNA was treated with gDNA wipe-out buffer (Qiagen) and reverse transcribed using the Quantitect reverse transcription kit (Qiagen) as per the manufacturer's instructions. For quantitative PCR ~20 ng of total cDNA was used per reaction and was run using FastStart SYBR Green reagent (Roche), cycle conditions were defined by the manufacturer's protocol. All qRT-PCR reactions were run using a Lightcycler 480-II. All primer sets used in this study were purchased from Qiagen (Table 1).

**Tissue fixation and immunohistochemistry**. Following euthanasia, livers were flushed through with 20 ml of physiological saline. Dissected liver lobes were then

fixed overnight in 10% neutral buffered formalin in PBS. Following fixation, tissue was dehydrated in 70% ethanol and processed to wax using a tissue processor. Four-micrometre tissue sections were cut and rehydrated through graduated alcohols. Tissue sections were antigen retrieved as per Table 2. For DAB staining of antigens, tissue sections were treated with 10% $H_2O_2$, washed, and then native streptavidin and biotin were blocked by using a commercial avidin and biotin blocking kit. Following this, sections were blocked with a universal protein block. Tissue sections were incubated overnight with primary antibodies diluted in antibody diluent. Following washing, biotin-conjugated secondary antibodies were incubated with tissue sections as per Table 2. Sections were further incubated with ABC vectorstain and colour was developed using DAB substrate. For tyramide-stained sections, the avidin/biotin step was omitted, and HRP-conjugated secondary antibodies and Alexa 488 or Alexa 594-conjugated tyramide was used as per the manufacturer's instructions. For directly conjugated immunofluorescent antibodies, the same protocol was performed, by omitting the $H_2O_2$ and Avidin/Biotin blocking stage. DAB-stained sections were imaged on an Olympus Dotslide, Hamamatzu Nanozoomer or an Olympus BX53 upright microscope. Immunofluorescence was imaged using a Nikon A1R confocal microscope.

**RNAScope**. RNAscope was performed on formalin-fixed liver tissue. Four-micrometre sections were used throughout, and all RNAScope performed in this study was done by Aquila Histoplex, Edinburgh, with the exception of Axin2, which was performed at the core histology facility, The Beatson Institute, Glasgow.

**Bile duct isolation**. To isolate bile ducts from both uninjured and injured livers, dissected liver was chopped into 5-mm³ pieces and digested in DMEM/F-12 media containing Collagenase-IV (Roche) and DNASe-I (Roche). Following digestion and dissociation, bile ducts become obvious as parenchyma is digested away. Bile ducts are strained through a 70-μm filter and extensively washed in PBS to remove any residual cells. Bile ducts are then used for downstream applications.

**Organoid culture**. Liver organoids were derived from isolated bile ducts. Briefly, isolated bile ducts were resuspended in 100% GFR Matrigel. Within 24 h, bile ducts form closed structures, and within 48 h budding can be seen from the duct. Following expansion, these ducts were removed from Matrigel by incubating with ice-cold Versene. Organoids were dissociated with pipetting and then re-plated in fresh 100% Matrigel. This process was repeated to expand organoids. The growth medium used in this study consisted of a base media of DMEM/F-12 supplemented with Glutamax, Penicillin/Streptomycin, Fungizone and HEPES. Just prior to feeding, the base media was supplemented with HGF, EGF, FGF10, Gastrin, Nicotinamide, N-Acetylcystine, B-27, Forskolin, Y-27632 (ROCK inhibitor), A83-01 (TGF-β inhibitor) and Chir99021 (GSK3β inhibitor).

To evaluate the effects of Vangl2 knockout on bile duct organoids (Vangl2[flox/flox]) versus control organoids (Vangl2[WT/WT]) Cre was expressed from the CMV promoter, in vitro, by infection of organoid structures with Lv-CMV-Cre (University of Edinburgh, SuRF facility) at an MOI of 5. For infection media was supplemented with 1/100 of Transdux reagent. For assays in which signalling was monitored in response to genotype, growth media was withdrawn from organoid cultures, and organoids were cultured in base media containing B-27, Forskolin, Nicotinamide and N-Acetylcystine alone for 48 h, at which point organoids were lysed for analysis.

The non-immortalised human cholangiocyte line (NHC-3 cells) was provided by Dr Jesus Banales (San Sebastián, Spain). Cells were grown in DMEM/F-12 supplemented with 5% FCS, 1× MEM non-essential amino acids, 1× MEM vitamin solution, 1× Lipid mixture (chemically defined), 1× MEM vitamin solution, 1× Penicillin/Streptomycin, 25 mg Soybean Trypsin Inhibitor, 1× Insulin/Transferrin/Selenium, 13.4 μg/ml Bovine Pituitary Extract, 3.93 μg/ml Dexamethasone, 3.4 μg/ml T3 (3,3,5-triiodo-L-thyronine), 0.025 μg/ml EGF and 4 μg/ml Forskolin, and were pH adjusted with 3.4 N NaOH.

**Preparation of bone marrow-derived macrophages**. Total bone marrow was isolated from adult mouse femurs and grown in DMEM/Ham's F-12 media containing glutamine, 10% FCS, Penicillin/Streptomycin and 50 ng/ml recombinant MCSF in low-attachment flasks. After 48 h, the concentration of MCSF was reduced to 25 ng/ml, and cells were maintained in this media for five further days. Following differentiation, bone marrow- derived macrophages (BMDMs) were plated at $1 \times 10^3$ cm$^{-2}$ and left to activate for 48 h, at which point they were used experimentally.

**FACS isolation of hepatic macrophages**. Dissected livers were minced and then digested for 2 h in DMEM/F-12 nutrient culture media containing 1% of FCS, 0.5 mg/ml Collagenase, 0.5 mg/ml Dispase and 0.1 mg/ml DNase1. Following digestion, liver tissue was mashed through a 70-μm cell strainer. Heptocytes and large cell clumps were removed from the cell mixture by slow speed centrifugation at $50 \times g$. The remaining cells were then pelleted and red cells lysed using Red Cell Lysis buffer (Gibco). Following lysis, cells were blocked in PBS containing BSA and sodium azide for 30 min on ice. Following blocking, cells were incubated in blocking buffer with CD45-PE and CD11b-PECy7 (Biolegend) antibodies for 1 h at room temperature and were then sorted using a FACS Aria II. Dead cells were excluded from sorting using DAPI.

**Table 1 qPCR primers used in this study.**

|  | Provider | Catalogue number |
|---|---|---|
| *Mouse* | | |
| Axin2 | Qiagen | QT00126539 |
| Ccnd1 | Qiagen | QT00154595 |
| Ctgf | Qiagen | QT00096131 |
| Lef1 | Qiagen | QT00148834 |
| Mmp7 | Qiagen | QT00110012 |
| Myc | Qiagen | QT00096194 |
| Ppia | Qiagen | QT00247709 |
| Ptk7 | Qiagen | QT00144032 |
| Vangl1 | Qiagen | QT00494235 |
| Vangl2 | Qiagen | QT00128422 |
| Wnt5a | Qiagen | QT00164500 |
| *Human* | | |
| PPIA | Qiagen | QT00052311 |
| VANGL1 | SAbiosciences | PPH21116A |
| VANGL2 | SAbiosciences | PPH12377B |
| WNT5A | SAbiosciences | PPH02410A |

**Table 2 Antibodies used in this study.**

| | Provider | Catalogue number | Conditions |
|---|---|---|---|
| *Primary antibodies* | | | |
| Alpha smooth muscle actin (1A4) | Sigma Aldrich | A2547 | 15 m Citrate. 1/2000 dilution O/N |
| β-actin | Cell Signalling Technologies | 8457P | 1/5000 O/N (western blot) |
| β-catenin | BD Transduction Laboratories | 610154 | 1 h Citrate 1/50 O/N |
| Calnexin | Enzo | ADI-SPA-860-D | 1/2000 O/N |
| CD45-PE | BioLegend | 103105 | 1 μl per 1 × 10e6 cells, 30 m RT |
| CD11b-PECy7 | BioLegend | 101215 | 1 μl per 1 × 10e6 cells, 30 m RT |
| CD68 (KP1) | Dako | M081401-2 | |
| c-JUN (total) (60A8) | Cell Signalling Technologies | 9165 | 30 m Citrate. 1/100 dilution O/N 1/1000 O/N (western blot) |
| Collagen-1 | Southern Biotech | 1310-01 | 5 m Citrate. 2/200 dilution 72 h |
| Ctgf | Abcam | ab6992 | No antigen retrieval 1/100 O/N |
| Desmin (Y66) | Abcam | Ab32362 | 15 m Citrate. 1/100 dilution O/N |
| E-cadherin DECMA-1 | Genetex | GTX11512 | 1/100 O/N |
| GFP | Abcam | ab13970 | 5 m Tris-EDTA. 1/500 O/N |
| Glutamine Synthatase | Abcam | ab64613 | 10 m Citrate 1/100 |
| Keratin-19 (TROMA-III) | Developmental Studies Hybridoma Bank | Troma-III | 10 m Citrate. 1/200 dilution O/N |
| pcJUN (ser73) (D47G9) | Cell Signalling Technologies | 3270 | 30 m Citrate. 1/100 dilution O/N |
| pcJUN (ser63) (54B3) (western blot) | Cell Signalling Technologies | 2361 | 1/1000 O/N |
| JNK (total) | Cell Signalling Technologies | 9252 | 30 m Citrate. 1/100 dilution O/N 1/1000 O/N (western blot) |
| pJNK (Thr183/Tyr185) (81E11) | Cell Signalling Technologies | 4668 | 30 m Citrate. 1/100 dilution O/N |
| PTK7 | Source Bioscience | LS-B10725 | 10 m Tris-EDTA 1/100 O/N |
| RFP | Abcam | ab62341 | 10 m Citrate. 1/100 dilution O/N |
| Vangl1 | Sigma Aldrich | HPA025235 | 10 m Tris-EDTA 1/100 O/N |
| Vangl2 | Sigma Aldrich | HPA027043 | 10 m Tris-EDTA 1/100 O/N |
| Vangl2 (western blot) | R&D Systems | AF4815 | 1/500 O/N |
| Wnt5a | LS Bio | LS-B4565 | 10 m Tris-EDTA 1/100 O/N |
| *Secondary antibodies* | | | |
| Anti-goat Alexa 594 | Thermo Fisher | A-27016 | 1/500 1 h RT |
| Anti-goat Biotinylated | Vector Laboratories | BA-5000 | 1/500 1 h RT |
| Anti-mouse Alexa 488 | Thermo Fisher | A32723 | 1/500 1 h RT |
| Anti-rabbit Alexa 594 | Thermo Fisher | A-21207 | 1/500 1 h RT |
| Anti-Rabbit Biotinylated | Vector Laboratories | BA-1000 | 1/500 1 h RT |
| Anti-rabbit HRP | Vector Laboratories | PI-1000 | 1/500 1 h RT |
| Anti-rat Alexa 488 | Thermo Fisher | A-21208 | 1/500 1 h RT |
| Anti-rat Alexa 594 | Thermo Fisher | A-21209 | 1/500 1 h RT |
| Tyramide Alexa 594 | Thermo Fisher | B40957 | 1/50 10 m RT |
| Tyramide Alexa 488 | Thermo Fisher | B40953 | 1/50 10 m RT |

**Western blotting**. Isolated bile ducts or organoids were lysed in RIPA buffer containing phosphatase (Thermo Fisher) and protease (MiniComplete, Roche) inhibitors. Protein quantification was determined using Pierce BCA reagent (Pierce) and quantified using a nanodrop. The standard curve for protein quantification was derived from the BCA reagent handbook using Albumin standards provided. In all, 20 μg of total protein was loaded onto a 4–12% NuPage Bis-Tris gel (Thermo Fisher). Prior to running, proteins were reduced with NuPage LDS sample buffer (4×) and NuPage Sample Reducing Agent (10×). All gels were run using NuPage MOPS SDS Running buffer containing NuPage Antioxidant. Proteins were transferred onto PVDF membrane (Amersham) using NuPage Transfer Buffer. Following transfer, membranes were blocked in either 5% BSA (Sigma Aldrich) in TBST or 5% dried milk (Marvel) in TBST. Membranes were incubated with primary antibodies (Table 2) at 4 °C overnight. Following washing with TBST, membranes were incubated with HRP-conjugated secondary antibodies (Table 2) at room temperature for 1 h. Following washing the signal was developed using ECL (Pierce). Uncropped images of gels are available in the source data file.

**Plasmid preparation**. Both the pT3-EF1aH N90-beta-catenin and PT2/C-Luc// PGK-SB13 plasmids were grown in LB media, containing 100 μg/ml ampicillin overnight. Cells were pelleted and plasmids were prepared with a Qiagen endotoxin-free maxiprep kit, as per the manufacturer's instructions. Plasmid concentration was determined using a Nanodrop.

**Image analysis**. Image analysis was conducted using ImageJ and macros written by Dr Tim J Kendall. Macros are available on request.

**Statistics**. All experimental groups were analysed for normality using a D'Agostino–Pearson Omnibus test. Groups that were normally distributed were compared with either a two-tailed Student's *t* test (for analysis of two groups) or using one-way ANOVA to compare multiple groups, with a post hoc correction for multiple testing. Non-parametric data were analysed using a Wilcoxon–Mann–Whitney U test when comparing two groups or a Kruskall–Wallis test when comparing multiple non-parametric data. Throughout $p < 0.05$ was considered significant. Data are represented as mean with S.E.M. for parametric data or median with S.D. for non-parametric data.

**Study approval**. All animal experiments were approved by the University of Edinburgh local ethics committee and were licensed by the UK Home Office. All patient material contained in this paper was approved by the NHS Lothian Bio resource ethics committee. No prospective tissue was collected in this study.

**Reporting summary**. Further information on research design is available in the Nature Research Reporting Summary linked to this article.

## Data availability

The authors declare that the data supporting the findings of this study are available within the paper and its supplementary information files. The source data underlying Figs. 1b, d, e, 2b–d, 3a, c–f, 4a, c, d, 5b–g and 6c, e–h and Supplementary Figs. 1a, e–g, 2a, 3a, f, 4a, d, e, h, 5a and 6b, c are provided as a Source Data file.

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

## Acknowledgements

The authors would like to thank Guoqiang Gu for originally providing the Krt19CreER[T] mouse line and Ed Boulter-Comer for technical reading of the paper. This work was funded by Primary Sclerosing Cholangitis (PSC) Support, The Alan Morement Memorial Foundation (AMMF, The Cholangiocarcinoma Charity) and core funding provided to the MRC Human Genetics Unit, by the Medical Research Council and the Wellcome Trust (207793/Z/17/Z). T.J.K. was partially funded by a Wellcome Trust Intermediate Clinical Fellowship (095898/Z/11/Z). O.S. and A.R. were funded by CRUK core funding to the CRUK Beatson Institute (A17196).

## Author contributions

D.H.W., E.J.J., R.P.M., M.L.W., S.H.W., P.T., A.T.R.N., N.T.Y. and A.R. performed experiments and analysed data. S.R.B. and D.O.B. bred, maintained and provided transgenic lines used in this study. S.W.O.D. and F.G.S. provided tissue and expertise in bile duct biology, they also provided sponsorship and financial support for A.T.R.N. P.C. provided advice and the Vangl2[GFP] knock-in mouse line through collaboration. C.H.D. collaborated on provision of the Vangl2[S464N] mouse line and Vangl2[S464N] tissues. D.J.H. generated and provided the Vangl2[flox/flox] mouse line. J.M.B. generated and provided the normal human cholangiocyte (NHC-3) primary cultures. O.J.S. provided RNAScope data and funding support for A.R. T.J.K. provided pathological support, wrote the macros for image analysis, contributed to the direction of the project and edited the paper. L.B. conceived and provided funding for the project, conducted experiments, analysed data, compiled, wrote the paper and led the project.

## Competing interests

The authors declare no competing interests.
