## [Peer Review File · Nature Communications]

Editorial Note: This manuscript has been previously reviewed at another journal that is not operating a transparent peer review scheme. This document only contains reviewer comments and rebuttal letters for versions considered at Nature Communications .

Reviewers' comments:

Reviewer #2 (Remarks to the Author):

This is a major revision of a manuscript examining the role of Wnt signaling through non- β -catenin pathways in biliary scarring.

The authors have a lot more work and importantly, removed a lot of extraneous material. This has markedly improved the manuscript and enhanced readability by the focus on Wnt5a and Vangl2. It's a clear and convincing example of a Wnt fibrosis response through a PCP pathway.

I have found only two typos to be fixed....

PicoSirius should be Picrosirius

Fibrillar should replace fibrilar

Reviewer #4 (Remarks to the Author):

In this manuscript, Wilson et al. assess the role of non-canonical Wnt signaling (Wnt/PCP pathway) in cholangiopathies. The authors show increased expression of both upstream (Wnt5a) and downstream targets (pJNK, p-cJUN, Ctgf, Mmp7) of the Wnt/PCP pathway in PSC patient tissues and in 2 different mouse models of biliary injury. They then demonstrate decreased fibrosis in these mouse models after inhibition of Wnt secretion, and after deletion of Vangl2, a scaffolding protein necessary for Wnt/PCP activity. This study therefore describes a heretofore-unknown role for non-canonical Wnt signaling in biliary epithelial cells which contributes to biliary fibrosis during cholestatic liver injury.

The strengths of this paper include use of multiple mouse biliary injury models and human tissues, analysis of both upstream and downstream pathway effectors, and employment of multiple readouts and parameters to ensure rigor and reproducibility. Although the findings are overall novel and comprehensive, and the authors demonstrate a good-faith effort to answer previous reviewer concerns, a few technical concerns remain:

It is important to demonstrate proof that macrophages (the source of Wnt5a) are taking up LGK974 after gavage. Loss of hepatocyte-specific glutamine synthetase expression is insufficient to show this, as the drug may have lower penetrance in non-parenchymal cells.

The authors should demonstrate efficient deletion of Wnt5a by LysMCre by isolating macrophages from WT and LysMCre::Wnt5aflox and performing qPCR for Wnt5a. Even isolated macrophages from baseline livers would be sufficient to prove the point.

Can the authors profile expression of all Wnts in both biliary injury models, to identify how expression (especially of Wnt5a) changes in earlier and later stages?

The authors should quantify %Krt19+ (or ROSA+ cells) in the Krt19CreERT::Vangl2flox model after biliary injury as has been done for the other models.

Minor:

In Figure 1a, c.v. (central vein) should be changed to p.v. (portal vein).

Ref. 13 (Okabe et al.) shows that BEC proliferation is Wnt-dependent but beta-catenin independent; this should be corrected in the Introduction.

Reviewer #5 (Remarks to the Author):

The authors have addressed majority of the concerns from the previous review cycle. However, as suggested by the reviewer, MMP7, which is known to activate CTGF, is restricted to the biliary epithelium potentially representing a localized mechanism by which CTGF is activated. If the authors want to link MMP to CTGF and activation of fibroblasts they should provide experimental evidence to knockdown of Mmp7 which is critical to support the central hypothesis.

Response to Reviewer Comments.

Title: Wnt-PCP signalling regulates scarring in biliary disease

Authors: Wilson DH, Jarman EJ, Mellin RP, Wilson ML, Waddell NT, Tsokkou P, Younger NT, Raven A, Bhalla SR, Noll ATR, Olde Damink SW, Schaap FG, Chen P, Bates DO, Banales JM, Dean CH, Henderson DJ, Sansom OJ, Kendall TJ, Boulter L.

Reviewer #2 (Remarks to the Author):

This is a major revision of a manuscript examining the role of Wnt signaling through non- β -catenin pathways in biliary scarring.

The authors have a lot more work and importantly, removed a lot of extraneous material. This has markedly improved the manuscript and enhanced readability by the focus on Wnt5a and Vangl2. It's a clear and convincing example of a Wnt fibrosis response through a PCP pathway.

I have found only two typos to be fixed....

PicoSirius should be Picosirius

Fibrillar should replace fibrilar

We would like to thank the reviewer for their comments and taking the time to review this manuscript for a second time. We have addressed the typographical errors detailed below.

Reviewer #4 (Remarks to the Author):

In this manuscript, Wilson et al. assess the role of non-canonical Wnt signaling (Wnt/PCP pathway) in cholangiopathies. The authors show increased expression of both upstream (Wnt5a) and downstream targets (pJNK, p-cJUN, Ctgf, Mmp7) of the Wnt/PCP pathway in PSC patient tissues and in 2 different mouse models of biliary injury. They then demonstrate decreased fibrosis in these mouse models after inhibition of Wnt secretion, and after deletion of Vangl2, a scaffolding protein necessary for Wnt/PCP activity. This study therefore describes a heretofore-unknown role for non-canonical Wnt signaling in biliary epithelial cells which contributes to biliary fibrosis during cholestatic liver injury.

The strengths of this paper include use of multiple mouse biliary injury models and human tissues, analysis of both upstream and downstream pathway effectors, and employment of multiple readouts and parameters to ensure rigor and reproducibility. Although the findings are overall novel and comprehensive, and the authors demonstrate a good-faith effort to answer previous reviewer concerns, a few technical concerns remain:

It is important to demonstrate proof that macrophages (the source of Wnt5a) are taking up LGK974 after gavage. Loss of hepatocyte-specific glutamine synthetase expression is insufficient to show this, as the drug may have lower penetrance in non-parenchymal cells.

Thank you for this point. In the manuscript, we are careful not to suggest that LGK974 would only act on macrophages, rather that this inhibitor suppresses the production of Wnt ligands from multiple cell lineages. To verify whether macrophages are the source of Wnt5a in this study, we then went on to delete Wnt5a specifically in macrophages using a LysMCre::Wnt5a^{flox/flox} transgenic mouse line. This gives us confidence that, even though our inhibitor approach could

suppress a range of Wnt ligands from different cells, Wnt5a from the myeloid lineage plays an important role in biliary scarring.

In order to address the reviewers specific concern, that LGK974 may have a lower penetrance in macrophages, we have given LGK974 or vehicle by gavage to mice with biliary disease. We have then isolated these macrophages by FACS for CD45+/CD11B+ cells and stained these cells for Wnt5a and the endoplasmic reticulum marker, Calnexin. LGK974 inhibits the addition of palmitate onto Wnt ligands and therefore inhibits their progression out of the endoplasmic reticulum and through the secretory pathway. In macrophages (N=677 individual cells) from LGK974 treated mice, we found co-localisation of Wnt5a and Calnexin in 50.51% of isolated macrophages. Whereas in macrophages isolated from vehicle treated animals (N=822), only 32.1% of macrophages demonstrated co-expression of Wnt5a and Calnexin, suggesting to us that in vivo, LGK974 does affect macrophages and results in the accumulation of Wnt5a in the ER, this data is now included in Supplementary Figure 3.

The authors should demonstrate efficient deletion of Wnt5a by LysMCre by isolating macrophages from WT and LysMCre::Wnt5a^{flox} and performing qPCR for Wnt5a. Even isolated macrophages from baseline livers would be sufficient to prove the point.

Thank you for this comment. We have shown using bone marrow derived macrophages and primary isolated macrophages from mice with biliary injury that Wnt5a is lost in the LysMCre::Wnt5a^{flox/flox} line and present the data in supplementary Figure 4. We have also updated the materials and methods to reflect our approach for isolation hepatic macrophages.

Can the authors profile expression of all Wnts in both biliary injury models, to identify how expression (especially of Wnt5a) changes in earlier and later stages?

Thank you for this point. We have included mRNA expression data for *Wnt5a* in figure 3a for both the DDC and TAA model. We do agree though that this only provides part of the picture and so we have used qRT-PCR arrays to look at the expression of all detectable Wnt ligands in ducts isolated from normal, DDC treated or TAA treated mice. We now include this in Supplementary Figure 2 and also in the text.

The authors should quantify %Krt19+ (or ROSA+ cells) in the Krt19CreERT::Vangl2^{flox} model after biliary injury as has been done for the other models.

Thank you for this point. These data are presented in Supplementary Figure 6 c and 6d; however, we had failed to cite this in the text. We have now clarified the reference to this data in the text (pg 15) and ensured that it is cited appropriately.

Minor:

In Figure 1a, c.v. (central vein) should be changed to p.v. (portal vein).

Thank you for spotting this typo. We have now changed the figure to reflect that this is a portal vein and not a central one.

Ref. 13 (Okabe et al.) shows that BEC proliferation is Wnt-dependent but beta-catenin independent; this should be corrected in the Introduction.

Apologies for the mis-citation of this paper. We have now moved it to a part of the text (highlighted) where it makes more sense with the narrative. This reference is now #15 on pg. 3.

Reviewer #5 (Remarks to the Author):

The authors have addressed majority of the concerns from the previous review cycle. However, as suggested by the reviewer, MMP7, which is known to activate CTGF, is restricted to the biliary epithelium potentially representing a localized mechanism by which CTGF is activated. If the authors want to link MMP to CTGF and activation of fibroblasts they should provide experimental evidence to knockdown of Mmp7 which is critical to support the central hypothesis.

Thank you for this point. We do not assess formally whether Ctgf is activated by Mmp7 in the context of biliary scarring. In our manuscript, we use transcription of these genes to read out pathway activation via cJun as they are known to be cJun target genes. We have been through this resubmission carefully to ensure that we do not inadvertently suggest that our data shows MMP7 dependant activation of CTGF.

REVIEWERS' COMMENTS:

Reviewer #4 (Remarks to the Author):

No further changes or additions are requested.